# Birth incidence, deaths and hospitalisations of children and young people with Down syndrome, 1990–2015: birth cohort study

Laura Anne Hughes-McCormack ,[1] Ruth McGowan ,[2] J P Pell,[3] Daniel Mackay,[3] Angela Henderson ,[1] Lisa O'Leary,[4] Sally-Ann Cooper [1]

¹Mental Health and Wellbeing research group, Institute of Health and Wellbeing, University of Glasgow, Glasgow, UK
²West of Scotland Regional Genetics Service, Queen Elizabeth University Hospital, Glasgow, UK
³Public Health, Institute of Health and Wellbeing, University of Glasgow, Glasgow, UK
⁴School of Health and Social Care, Edinburgh Napier University, Edinburgh, UK

**Correspondence to**
Professor Sally-Ann Cooper;
Sally-Ann.Cooper@glasgow.ac.uk

## ABSTRACT

**Objective** To investigate current Down syndrome live birth and death rates, and childhood hospitalisations, compared with peers.

**Setting** General community.

**Participants** All live births with Down syndrome, 1990–2015, identified via Scottish regional cytogenetic laboratories, each age–sex–neighbourhood deprivation matched with five non-Down syndrome controls. Record linkage to Scotland's hospital admissions and death data.

**Primary outcome** HRs comparing risk of first hospitalisation (any and emergency), readmission for children with Down syndrome and matched controls were calculated using stratified Cox proportional hazards (PH) model, and length of hospital stay was calculated using a conditional log-linear regression model.

**Results** 689/1479 (46.6%) female and 769/1479 (51.9%) male children/young people with Down syndrome were identified (1.0/1000 births, with no reduction over time); 1235 were matched. 92/1235 (7.4%) died during the period, 18.5 times more than controls. More of the Down syndrome group had at least one admission (incidence rate ratio(IRR) 72.89 (68.72–77.32) vs 40.51 (39.15–41.92); adjusted HR=1.84 (1.68, 2.01)) and readmissions (IRR 54.85 (51.46–58.46) vs 15.06 (14.36–15.80); adjusted HR=2.56 (2.08, 3.14)). More of their admissions were emergencies (IRR 56.78 (53.13–60.72) vs 28.88 (27.73–30.07); first emergency admission adjusted HR=2.87 (2.61, 3.15)). Children with Down syndrome had 28% longer first admission after birth. Admission rate increased from 1990–2003 to 2004–2014 for the Down syndrome group (from 90.7% to 92.2%) and decreased for controls (from 63.3% to 44.8%).

**Conclusions** We provide contemporaneous statistics on the live birth rate of babies with Down syndrome, and their childhood death rate. They require more hospital admissions, readmissions emergency admissions and longer lengths of stays than their peers, which has received scant research attention in the past. This demonstrates the importance of statutory planning as well as informal support to families to avoid added problems in child development and family bonding over and above that brought by the intellectual disabilities associated with Down syndrome.

## Strengths and limitations of this study

► A whole country population-based study on Down syndrome, with comprehensive identification via all NHS Scottish Regional Genetics Centres.
► 25-Year period of linked data.
► Account taken of the potential confounders of age, sex and extent of neighbourhood deprivation; underlying conditions were not investigated.
► Some of the Down syndrome individuals could not be matched, mostly due to missing data on postcode, and so were omitted from analyses.
► The use of routinely collected hospital and death statistics has many advantages including large-scale coverage, but there is likely to be a degree of coding inaccuracy which is not quantified.

## INTRODUCTION

Improvements in health and social care and attitudes to disability have increased the survival of people with Down syndrome in recent decades.[1–6] In England and Wales, it was estimated that there were 37 090 people with Down syndrome in 2011,[7] but exact numbers are not reported. The National Down Syndrome Cytogenetic Register for England and Wales produces data on the estimated live birth rate for Down syndrome, using assumptions on termination rates for those receiving a prenatal diagnosis. In 2013, it reported an estimated live birth rate of 1.0/1000.[8] In the USA, de Graaf *et al*[9] estimated live birth and population prevalence for Down syndrome for all nine states in which data were available. They reported a live birth prevalence rate of 12.1/10 000, and an increase in the number of people living with Down syndrome from 1950 to 2010. In Canada, the live birth rate for Down syndrome was reported to have remained stable between 2005 and 2013, and higher than in the UK and USA rates, at approximately 13.5/10 000.[10] Similar data are

not available for Scotland. Given the changing population demographic for people with Down syndrome, and limited evidence base, we require a better understanding of their live birth rate and future health service needs.[4]

Down syndrome is associated with a range of congenital anomalies, particularly cardiac, and a higher risk for respiratory, immunological, endocrine and gastrointestinal conditions.[4 11] This phenotype puts people with Down syndrome at risk of hospitalisation. This may impact on their development and family relationships, and so is an added disadvantage on top of their intellectual disabilities. Despite this, few previous studies have investigated hospitalisation of children and young people with Down syndrome, compared with the general population. Fitzgerald *et al*[12] studied hospital admissions of 405 children with Down syndrome born in Western Australia between 1983 and 2004. They found that the children with Down syndrome were hospitalised five times more often when compared with previously published general population data for the single year of 1995. They had no general population comparison data for length of stay. Zhu *et al*[13] investigated hospitalisations of 3212 children and adults with Down syndrome compared with 67 204 children and adults without Down syndrome in Denmark, between 1977 and 2008. The people with Down syndrome had more than twice the rate of hospital admissions, and nearly three times as many bed days. Both studies found that length of admissions of the people with Down syndrome had reduced over the periods studied, and Zhu *et al*[13] reported more bed days for males, children under 5 years, and those with congenital heart anomalies. Other studies have lacked general population comparison data or been limited to single conditions. We are not aware of any large-scale studies specifically of children and young people with Down syndrome investigating these issues compared with their peers. There has been very little other investigation of the factors that influence length of hospital stay of children and young people with Down syndrome, and we located no studies on risk of readmission or on emergency admissions of children and young people.

Without accurate information on live birth and death rates, and hospital admission needs, it is not possible to plan for the healthcare support that children with Down syndrome and their families need.

This study's aims were to investigate the (1) incidence of live births of people with Down syndrome over a 25-year period, (2) frequency of deaths of children and young people with Down syndrome compared with matched controls and (3) hospital admission frequency and duration, emergency admissions and readmissions of children and young people with Down syndrome compared with matched controls.

## METHODS

Written informed consent was not obtained from participants for their clinical records to be used in this study, but patient records were anonymised and de-identified prior to analysis. Safehaven approval was granted to approved researchers to analyse the data.

### Patient and public involvement

This study was undertaken by the Scottish Learning Disabilities Observatory, which has a specific remit for people with intellectual disabilities, including people with Down syndrome; its steering group includes partners from third sector organisations, including Down Syndrome Scotland. Results from this study will be disseminated for people with Down syndrome and their families/carers in easy-read version via the Scottish Learning Disabilities Observatory website and newsletters.

### Study sample, setting and process

All NHS Scottish Regional Genetics Centres (the east, north, south-east and west Scotland centres) identified all live birth infants screened positive for Down syndrome postnatally (live births with trisomy 21, mosaic trisomy 21 or unbalanced translocation resulting in trisomy 21) between 1 January 1990 and 1 December 2015. In Scotland, everyone is given a unique Community Health Index (CHI) at birth which is used in their health records. The CHI database is held centrally by National Services Scotland (NSS), and was used to identify five controls without Down syndrome for each person with Down syndrome, matched on sex, age (by month and year of birth) and neighbourhood deprivation (using the Scottish Index of Multiple Deprivation (SIMD)).[14 15] Down syndrome infants who could not be matched due to missing data or had less than 1 year of potential follow-up time (ie, births in 2015) were excluded from statisitcal analyses. The CHI also provides a means to record link each person identified with Down syndrome and their matched controls to routinely collected hospital admissions data (Scottish Morbidity Records 01: SMR01),[16] and National Records of Scotland death certificate data.[17] The end of the period of follow-up was 1 December 2015. However, children born in 2015 were excluded from the statistical analyses to allow a minimum potential follow-up time of 1 year for all children.

### Data sources and definitions
#### Live births
Live births of babies with Down syndrome from all NHS Scottish Regional Genetics Centres, 1990–2015; Scottish live births in the whole population, from the National Records of Scotland births time series, 1990–2015 (NRS, 2019).[17]

#### Death
Deaths and causes of death by the International Classification of Diseases codes[18] according to death certificates registered at National Records of Scotland.

#### Hospital admissions
SMR01 contains episode-based records for all non-psychiatric, non-obstetric acute hospital admissions in

Scotland. The information collected includes the date of admission to hospital and discharge from it and whether the admission was routine or emergency. In SMR01, continuous periods of care are accounted for with a continuous inpatient stay (CIS) marker. This CIS marker ensures that a series of individual episodes over an unbroken period of care (eg, transfers between wards or hospitals) can be identified and treated as one admission rather than several admissions. Transfer of a baby to a neonatal intensive care unit after birth is counted as a first admission. Data quality assurance assessments for SMR01 are performed periodically.[19 20]

### Admission type
Admission type is coded as emergency or routine. 'Emergency' admissions are those which were unplanned in advance; planned admissions are 'routine' admissions.

### Birth year group
Grouped into the most recent cohort (2004–2014) and the least recent cohort (1990–2003).

### Scottish Index of Multiple Deprivation
SIMD matching with controls and reporting is by quintiles, where SIMD 1 is the most deprived neighbourhoods and SIMD 5 is the most affluent neighbourhoods. SIMD is calculated at datazone level, identified from postcodes.

### Discharge type
Discharge type is coded as a regular discharge or irregular discharge. Irregular discharges include, for example, a patient discharging himself/herself against medical advice, or death.

### All follow-up/censoring
Children were followed up from birth, and all models were censored on death or 1 December 2015 (whichever came first) unless stated otherwise.

## Analyses
### Incidence of Down syndrome
Incidence of Down syndrome births was calculated for each calendar year from 1990 to 2015 inclusive.

### Death
Incidence rates for mortality per person time (per 1000) were described, split by age group (for all ages, 0–1 month of life, from 1 to 12 months, from 13 to 60 months, from 61 to 120 months, from 121 to 180 months and from 181 to 240 months), as were underlying causes of death, for people with Down syndrome and their matched controls.

### First hospital admissions
Incidence rates for first admissions per person time (per 1000) for people with Down syndrome compared with controls were described, split by age group (for all ages, 0–1 month, from 1 to 12 months and from 13+ months). Descriptive data for first admissions were presented including by sex, SIMD, year of birth and duration of first admissions in days, and this was compared using

dependent t-tests for continuous variables and $\chi^2$ tests for categorical variables or Mann-Whitney U tests for length of hospital stay. The relative risk of first hospital admission was compared for people with Down syndrome and controls using stratified (by sex, birth year group, SIMD) Cox regression models (log–log plots confirmed the proportional hazards assumption was met). The follow-up period was defined as from date of birth until date of first admission, and admission type was entered as a potential confounder.

### Risk of emergency hospital admission
The relative risk of emergency first hospital admission was compared for people with Down syndrome and controls, analysed using stratified (by sex, birth year group, SIMD) Cox regression models. The follow-up period was defined as from date of birth until date of first admission.

### Duration of hospital admission
Conditional (stratified by sex, birth year group, SIMD) linear regression was used to model the duration of first hospital admission (log transformed to help negate potential skewing of results) as the outcome, comparing Down syndrome and control groups. Admission type was entered as a potential confounder.

### Risk of readmission
The relative risk of readmission (the next admission after the first admission) was compared for people with Down syndrome and controls, analysed using stratified (by sex, birth year group, SIMD) Cox regression models. The follow-up period was defined as date of discharge from the first admission until date of the next admission. First admission type and discharge type from first admission were entered as potential confounders, and the interaction of group (Down syndrome vs controls)*admission type (emergency vs routine) was included.

Reference groups for all regression analyses were: not having Down syndrome, females, most affluent neighbourhoods, routine admissions and regular type of hospital discharges. All analyses were conducted with IBM SPSS V.22.

## RESULTS
Between 1 January 1990 and 1 December 2015, the Scottish Regional Genetics Centres identified 1479 live births with Down syndrome. One thousand two hundred and thirty-five people were matched with 6175 controls. One hundred eighty-seven (12.6%) infants with Down syndrome were excluded from statistical analyses, as they could not be matched due to missing data on SIMD quintile (n=160) in view of missing postcode information, sex (n=21), month and year of birth (n=<5) and CHI for example, due to migration out of Scotland (n=<5). A further 57 children were excluded as they were born in 2015 (and had less than 1 year of potential follow-up time). Of the 1479 infants born with Down syndrome, 689

**Table 1** Incidence of Down syndrome by year of birth

| Year of birth | Number of births registered in Scotland (all people) n | Down syndrome n (%) |
|---|---|---|
| 1990 | 65 973 | 60 (0.09) |
| 1991 | 67 024 | 66 (0.10) |
| 1992 | 65 789 | 38 (0.06) |
| 1993 | 63 337 | 35 (0.06) |
| 1994 | 61 656 | 43 (0.07) |
| 1995 | 60 051 | 46 (0.08) |
| 1996 | 59 296 | 40 (0.07) |
| 1997 | 59 440 | 72 (0.12) |
| 1998 | 57 319 | 54 (0.09) |
| 1999 | 55 147 | 52 (0.09) |
| 2000 | 53 076 | 60 (0.11) |
| 2001 | 52 527 | 61 (0.12) |
| 2002 | 51 270 | 44 (0.09) |
| 2003 | 52 432 | 58 (0.11) |
| 2004 | 53 957 | 79 (0.15) |
| 2005 | 54 386 | 63 (0.12) |
| 2006 | 55 690 | 44 (0.08) |
| 2007 | 57 781 | 70 (0.12) |
| 2008 | 60 041 | 63 (0.10) |
| 2009 | 59 046 | 73 (0.12) |
| 2010 | 58 791 | 59 (0.10) |
| 2011 | 58 590 | 63 (0.11) |
| 2012 | 58 027 | 59 (0.10) |
| 2013 | 56 014 | 58 (0.10) |
| 2014 | 56 725 | 62 (0.11) |
| 2015 | 55 098 | 57 (0.11)* |
| Total | 1 508 483 | 1479 (0.10) |

*In 2015, the Down syndrome total is for 11 months only, so the % in 2015 is adjusted to 11/12 months in the general population.

(46.6%) were girls, 769 (51.9%) were boys and sex was not recorded for 21 (1.4%). The proportion of births for the Down syndrome population by the five SIMD quintiles were: 1=21.3%, 2=20.4%, 3=18.5%, 4=20.1% and 5=19.7%, and SIMD data were missing for 160 (10.8%). Of the 1235 infants with Down syndrome with matched controls, 591 (47.9%) were girls and 644 (52.1%) were boys. The proportion of births for the Down syndrome population, by the five SIMD quintiles were: 1=21.0%, 2=20.0%, 3=18.2%, 4=19.8% and 5=19.9%. Hence they appear to be representative of all the Down syndrome infants that were born.

### Incidence of Down syndrome live births over time
Given the total number of infants born was 1 508 483, the incidence of Down syndrome live births was 1479/1

508 483 (1.0/1000 births). Table 1 and figure 1 show the incidence of Down syndrome live births for each year of the study period (with a moving average smoothed trend line to account for noise from year on year variation). The birth rate in Scotland has fallen since the early 1990s, while the incidence of Down syndrome shows some year-to-year variation and appears to have remained the same or risen since the early 1990s.

### Deaths
Of the 1235 people with Down syndrome with matched controls, 92 (7.4%) died during the study period (165 744 person months of follow-up for the Down syndrome group), of whom 47 (51.0%) were women and 45 (48.9%) were men. Of the 6175 matched controls, 23 (0.4%) died during the period (895 776 person months follow-up for the controls), of whom 9 (39.1%) were women and 14 (60.9%) were men. Death was therefore 18.5 times more common in the Down syndrome group. Most of the deaths of the people with Down syndrome occurred in infancy. Death incidence rates per person time (per 1000) by age groups for the Down syndrome population versus controls were 0.56 (95% CI 0.45 to 0.68) (n=92) vs 0.03 (95% CI 0.02 to 0.04) (n=23) for all ages, 17.81 (95% CI 9.87 to 32.17) (n=11) vs 0 for the first month of life, 3.87 (95% CI 2.96 to 5.07) (n=53) vs 0.17 (95% CI 0.09 to 0.29) (n=12) from 1 to 12 months, 0.36 (95% CI 2.96 to 5.07) (n=18) vs 0.02 (95% CI 0.01 to 0.05) (n=6) from 13 to 60 months, 0.15 (95% CI 0.07 to 0.32) (n=7) vs 0.01 (95% CI 0.00 to 0.04) (n=<5) from 61 to 120 months, 0.09 (95% CI 0.03 to 0.31) (n=<5) vs 0 from 121 to 180 months and 0 vs 0.02 (95% CI 0.00 to 0.08) (n=<5) from 181 to 240 months. The most common certified underlying causes of death for the Down syndrome group were congenital heart anomalies (n=33, 34.4%), Down syndrome (n=26, 27.1%) and infection (excludes respiratory infections) (n=11, 11.5%). This pattern was different compared with controls in whom causes of deaths were due to a range of other factors not found for people with Down syndrome (n=8, 34.8%), prematurity/perinatal causes (n=5, 21.7%) and congenital heart anomalies (n=5, 21.7%). Although less common than some other causes, leukaemia was a cause of death among people with Down syndrome (n=<5) compared with controls where no deaths were from leukaemia.

### Admissions
Table 2 presents details on the first all-cause hospital admissions for the two groups. One thousand one hundred and five people out of 1235 (89.5%) of the Down syndrome group had at least one hospital admission compared with 3305/6175 (53.5%) of the control group. Admission incidence rates per person time (per 1000) by age groups for the Down syndrome population versus controls were 72.89 (95% CI 68.72 to 77.32) (n=1105) vs 40.51 (95% CI 39.15 to 41.92) (n=3305) for all ages, 38.87 (95% CI 26.05 to 57.99) (n=24) vs 13.29 (95% CI 9.79 to 18.06) (n=41) for the first month of life, 56.44 (95% CI 52.03 to

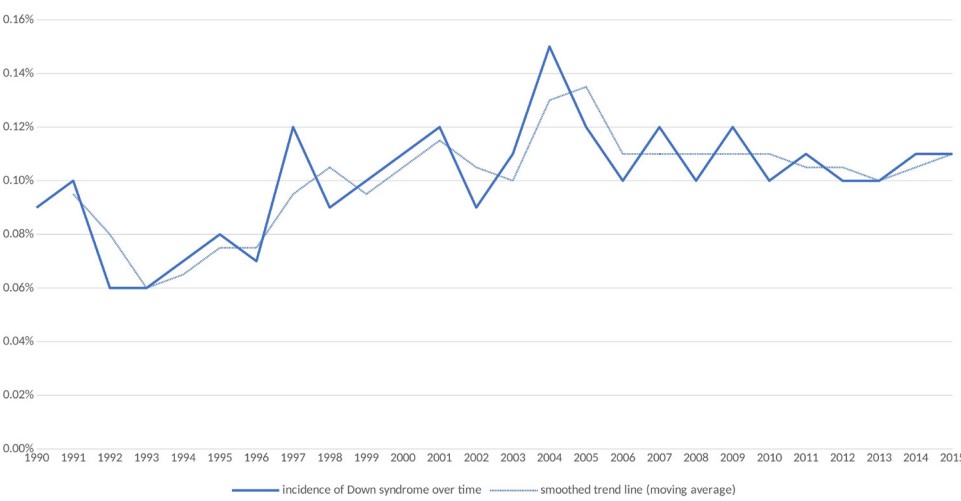

**Figure 1** Incidence of Down syndrome births by year in Scotland from 1990 to 2015.

61.23) (n=580) vs 24.94 (95% CI 23.62 to 26.33) (n=1298) from 1 to 12 months, 117.47 (95% CI 107.62 to 128.22) (n=501) vs 74.55 (95% CI 71.32 to 77.92) (n=1966) from 13+ months. Compared with the control group, the Down syndrome group were younger at the time of first hospital admission (median=0 months (IQR 0–300 months) vs 24 months (IQR 0–288)), had less admissions among females, were more equally spread across neighbourhoods while the control group had a gradient across SIMD groups with those in the most deprived areas having more hospital admissions, and had longer duration of admissions (5.03 days (median=2.00; IQR=1.00–4.00), vs 1.78 days (median=1; IQR=1.00–1.00)) by 3.25 days. The rate of admissions slightly increased over time for people with Down syndrome (from 528, 90.7% (in 1990–2003) to 577,

92.2% (in 2004–2014)) and decreased for controls (from 1837, 63.3% (in 1990–2003) to 1468, 44.8% (in 2004–2014)). The gap between the higher rate of admissions has widened over time for people with Down syndrome compared with controls.

Emergency admission incidence rates per person time (per 1000) by age groups for the Down syndrome population versus controls were 56.78 (95% CI 53.13 to 60.72) (n=861) vs 28.88 (95% CI 27.73 to 30.07) (n=2352) for all ages, 27.53 (95% CI 17.12 to 44.23) (n=17) vs 11.03 (95% CI 7.88 to 15.43) (n=34) for the first month of life, 48.66 (95% CI 44.57 to 53.11) (n=500) vs 20.42 (95% CI 19.23 to 21.69) (n=1063) from 1 to 12 months, 80.66 (95% CI 72.57 to 89.65) (n=344) vs 47.71 (95% CI 45.14 to 50.42) (n=1255) from 13+ months. Table 3 displays the results

| Table 2 First all-cause hospital admissions characteristics | | | |
|---|---|---|---|
| | **Down syndrome (n=1105)** | **Control group (n=3305)** | |
| | **n(%), Median (IQR)** | **n(%), Median (IQR)** | **P value** |
| Age at admission (months) | 0 (IQR=0–300) | 24 (IQR=0–288) | <0.001 |
| Sex | | | |
| Male | 572 (51.8%) | 1840 (55.7%) | <0.001 |
| Female | 533 (48.2%) | 1465 (44.3%) | |
| Birth year group | | | |
| Most recent | 577 (92.2%) | 1468 (44.8%) | <0.001 |
| Less recent | 528 (90.7%) | 1837 (63.3%) | |
| SIMD | | | |
| Most affluent (5) | 217 (19.6%) | 570 (17.2%) | <0.001 |
| 4 | 226 (20.4%) | 609 (18.4%) | |
| 3 | 203 (18.4%) | 591 (17.9%) | |
| 2 | 230 (20.8%) | 720 (21.8%) | |
| Most deprived (1) | 229 (20.7%) | 814 (24.6%) | |
| Duration of admission (days) | 2.00 (IQR=1.00–4.00) | 1.00 (IQR=1.00–1.00) | <0.001 |

Denominator for birth year group most recent/less recent are for Down syndrome n=653/582/control group n=3275/2900.
SIMD, Scottish Index of Multiple Deprivation.

**Table 3** Risk for first all cause hospital admission

| | Separately entered | | Simultaneously entered | |
|---|---|---|---|---|
| | **HR value** | **95% CI** | **HR value** | **95% CI** |
| Group | | | | |
| No Down syndrome | ref | ref | ref | ref |
| Down syndrome | 2.74† | 2.52 to 2.98 | 1.84† | 1.68 to 2.01 |
| First admission type | | | | |
| Routine | ref | ref | ref | ref |
| Emergency | 1.10* | 1.00 to 1.20 | 1.19† | 1.08 to 1.30 |

Stratified models by sex, birth year group, SIMD.
*P<0.05.
†P<0.001.
SIMD, Scottish Index of Multiple Deprivation.

of the Cox regressions for time to first all-cause hospital admission. The risk of admission was higher among people with Down syndrome compared with controls, with an unadjusted HR=2.74 (95% CI 2.52 to 2.98). This remained after adjusting for type of first admission: HR=1.84 (95% CI 1.68 to 2.01).

Table 4 displays the results of the Cox regressions for time to first emergency hospital admission. The risk of emergency admission was higher among people with Down syndrome compared with controls, with a HR=2.87 (95% CI 2.61 to 3.15).

## Duration of hospitalisation

The duration of first hospital admission was significantly longer for the Down syndrome group at 5.03 days (median=2 days) versus 1.78 days (median=1 day) by 3.25 days. After adjustment (with a log transformed time variable, to help account for any data skew) for sex, birth year group, SIMD and type of admission, the Coefficient=0.25 (95% CI 0.22 to 0.28) (table 5). This log transformed coefficient can be interpreted as a ratio of difference for the Down syndrome population compared with the control group (exponentiate of the log transformed coefficient, $\exp(0.248)=1.28145993219-1 \times 100=28$, with a baseline length of stay (intercept) $\exp(0.12)=1.12749685158$). This means the average length of stay was 28% higher in the Down syndrome population, compared with the controls. All exponentiated values are presented as percentages in table 5.

**Table 4** Risk for first emergency hospital admission

| | Separately entered | |
|---|---|---|
| | **HR value** | **95% CI** |
| Group | | |
| No Down syndrome | ref | ref |
| Down syndrome | 2.87* | 2.61 to 3.15 |

Stratified models by sex, birth year group, SIMD.
*P<0.001.

## Risk of readmission

Of the Down syndrome group, 945/1105 (85.5%) had at least one readmission, compared with 1685/3305 (51.0%) of the control group. Readmission incidence rates per person time (per 1000) for the Down syndrome population versus controls were 54.85 (95% CI 51.46 to 58.46) (n=945) vs 15.06 (95% CI 14.36 to 15.80) (n=1695). The baseline age at readmission was lower for the Down syndrome group with a median age of 6 months (IQR 1–17) compared with the controls with a median age of 11 months (IQR 5–20; p<0.001). The unadjusted HR of Down syndrome for time to readmission following discharge from first admission was 2.44 (95% CI 2.13 to 2.79). The HR when adjusted for type of admission, type of first discharge and group*emergency interaction was 2.56 (95% CI 2.08 to 3.14). Results are displayed in table 6.

## DISCUSSION
### Principal findings and interpretation
We found the Scottish incidence of Down syndrome live births to be 1.0/1000 births over the last 25 years, with it being possibly higher now than in the early 1990s. The Scottish birth rate has fallen overall, but incidence of Down syndrome has not. Death rate was 18.5 times higher for the children/young people with Down syndrome than for their controls, and their causes of death differed, including causes that might be expected to have been amenable to good healthcare such as infections.

Over a 25-year period, we found that children/young people with Down syndrome compared with matched controls were considerably more likely to have had at least one hospital admission (IRR of 72.89 compared with 40.51; adjusted HR=1.84) and readmissions (IRR of 54.85 compared with 15.06; adjusted HR=2.56). Their admissions were more likely to have been emergency admissions (IRR of 56.78 compared with 28.88; first emergency admission HR=2.87) and of longer duration 28% longer first admission after birth. Given the higher rate of first emergency admissions in the Down syndrome group, we

**Table 5** Duration of first all-cause hospital admission (log transformed time variable)

|  | Separately entered | | | Simultaneously entered | | |
|---|---|---|---|---|---|---|
|  | Coef | 95% CI | Exp % | Coef | 95% CI | Exp % |
| Group | | | | | | |
| No Down syndrome | ref | ref | 27 | ref | ref | 28 |
| Down syndrome | 0.24* | 0.21 to 0.27 | | 0.25* | 0.22 to 0.28 | |
| First admission type | | | | | | |
| Routine | ref | ref | 2 | ref | ref | 4 |
| Emergency | 0.02 | 0.00 to 0.04 | | 0.04* | 0.02 to 0.06 | |

Stratified models by sex, birth year group, SIMD.
*P<0.001.
Exp %, exponentiated values as percentages; Coef, Coefficient.

included the interaction of group×type of first admission (emergency or routine) in the analysis of readmissions, but they still remained more likely to be readmitted. We are not aware of any previous study of readmissions and emergency admissions in this population. The rate of admissions slightly increased over time (from 1990–2003 to 2004–2014) for the children/young people with Down syndrome (from 90.7% to 92.2%), while it decreased for the control group (from 63.3% to 44.8%), perhaps reflecting changes in medical practice over time, to keep children out of hospital as far as possible, but admitting for serious problems which the Down syndrome children may be more likely to experience. These findings are important, as they demonstrate the impact of additional health needs to both children and young people with Down syndrome, and to their families. This demonstrates the importance of statutory as well as informal support to families to avoid added problems in child development over and above the intellectual disabilities

that the children with Down syndrome experience, and the impact caring can have on families. Additionally, the high rates of emergency admissions and readmissions may indicate some poorer quality management of healthcare at the primary care level (to avoid admissions), and have not to our knowledge been reported before, though other factors, including underlying health conditions may contribute.

The Down syndrome group were younger at the time of first hospital admission, had a more equal sex distribution for first admission than the control group and an equal distribution across extent of neighbourhood deprivation, while the control group had a gradient with those in the most deprived areas more likely to have had a hospital admission. This suggests that at present, the children with Down syndrome appear to receive comparable care in terms of first admission, regardless of their neighbourhood, and highlights that findings in the general population cannot be relied on when planning services

**Table 6** Risk of readmission following discharge from first admission

|  | Separately entered | | Simultaneously entered | |
|---|---|---|---|---|
|  | HR value | 95% CI | HR value | 95% CI |
| Group | | | | |
| No Down syndrome | ref | ref | ref | ref |
| Down syndrome | 2.44† | 2.13 to 2.79 | 2.56† | 2.08 to 3.14 |
| First admission type | | | | |
| Routine | ref | ref | ref | ref |
| Emergency | 0.97* | 0.83 to 1.13 | 1.39 | 1.05 to 1.84 |
| Discharge type | | | | |
| Routine | ref | ref | ref | ref |
| Irregular | 0.86 | 0.83 to 0.89 | 0.85 | 0.82 to 0.88 |
| Group* first admission type | | | | |
| – | – | – | 1.06 | 0.78 to 1.44 |

Stratified models by sex, birth year group, SIMD.
*P<.05.
†P<0.001.

for people with Down syndrome/intellectual disabilities, such as focussing services in areas of greatest deprivation. Families of children and young people with Down syndrome need focused support regardless of the areas they reside in.

## Comparison with previous literature

As well as being 18.5 times more likely to die than their controls, we found that the children/young people with Down syndrome were more likely to be hospitalised, emergency hospitalised, readmitted and for longer durations. We have not identified any previous studies on readmission with which to compare our findings, nor on emergency admissions. For the earlier period of 1977–2008 in Denmark, Zhu *et al*[13] reported that children and adults with Down syndrome had more than twice the rate of hospital admissions (all admissions, not separately reported for first admissions or repeat admissions), similar to our adjusted HRs for first and repeat admissions, but we specifically investigated children and young people unlike that study. The smaller Australian study for the earlier period of 1983–2004 suggested that the hospitalisation rate was five times higher, but relied on previously reported general population data from 1995.[12]

As our study shows, and as previously reported in a study of children with Down syndrome identified from population-based sources in 1997 (n=210) and 2004 (n=208) in Western Australia,[21] it is highly likely that admission rates change over time, so these two studies are not strictly comparable to ours, and we have found no studies contemporaneous with ours with which to draw comparisons. Previously, small-scale studies reported length of admissions of children with Down syndrome to be 4.7 days in a Swedish study of 211 children followed up for at least 17 years,[22] and 7 days specifically for bronchiolitis in a small retrospective case–control study in Singapore,[23] compared with the 5.0 days we found. A further small study of 213 children with Down syndrome aged 0–3 years in America reported high admission rates.[24]

The live birth incidence for Down syndrome we report is similar to that recently reported in England, USA and Canada.[8–10] Antenatal screening changes may underlie the variability in the birth rates found over time. Previous research in Scotland showed that decisions to terminate pregnancy following antenatal diagnosis of aneuploidy had fallen over time,[1] which our results are also possibly suggestive of from the early 1990s to more recently. In contrast, in some other countries (eg, Denmark), following the introduction of antenatal combined screening for Down syndrome in 2004, there was a concurrent increase in prenatal diagnoses and the number of Down syndrome live births decreased suddenly and significantly before then stabilising.[25] A higher proportion of the Scottish population have catholic faith compared with Denmark which might contribute in part to differences between countries: a range of complex factors influence women's decision making regarding prenatal screening and diagnosis of Down syndrome, and it is possible that more

societal acceptance of diversity and people with disabilities, or better availability of family support may influence decisions in Scotland over time,[1 26] although this requires further investigation.

## Strengths and limitations

This is the first population-based study examining incidence of Down syndrome and healthcare utilisation over time in Scotland compared with the general population, and the first worldwide to investigate emergency hospital admissions and readmissions. It controlled for the potential confounders of age, sex and SIMD. It was not subject to selection bias, as the population identification was based on routine data sources from all genetic service laboratories in Scotland (ie, the whole population). The Cox regression analysis for risk of admissions and readmissions provides advantages over logistic or linear techniques as this includes consideration of time to event (person time at risk). However, there were some missing control data, mostly due to missing postcodes, and this is a limitation of the study. There were no data on ethnicity which is a limitation; the Scottish population is 96% white. The use of routinely collected hospital and death statistics has many advantages including large-scale coverage, but there is likely to be a degree of coding inaccuracy which is not quantified. Children with Down syndrome have higher risk of comorbidities, so may be more likely to be transferred to neonatal intensive care units which might explain the increased risk of first hospital admission rates, but we could not distinguish such admissions in our dataset. We did not aim to investigate underlying conditions which are associated with admissions and death in the two groups.

## CONCLUSIONS

As people with Down syndrome are living longer than in the past, we have provided contemporaneous statistics for the live birth rate of babies with Down syndrome and their childhood death rate compared with peers. We have found that their increasing survival is not without health consequences for the child, their family and health services—we report the majority require hospital admissions, readmissions and longer lengths of stays than their peers, which has received scant research attention in the past. Some factors may even suggest poorer healthcare than their peers, such as higher rates of emergency admissions, although we did not aim to investigate the conditions associated with admission. Together this brings important new findings necessary to address the challenge of planning support that children with Down syndrome and their families need. They require adequate healthcare, and also an awareness of the impact that hospitalisation can have on child development and family bonding. There may be a need to develop tailored packages of support for families with children/young people with Down syndrome to cope with the higher healthcare needs of their offspring.

**Contributors** LAH-M contributed to the study design, the analysis and interpretation of data, drafted the first article and revised it critically; DM, AH and LO contributed to the study design, interpretation of data and revised the article critically; S-AC, JPP and RM conceived and designed the study, contributed to study analysis, interpretation of data and revised the article critically. All authors approved the final version.

**Funding** MRC Mental Health Data Pathfinder Award (MC_PC_17217), and the Scottish Government via the Scottish Learning Disabilities Observatory.

**Competing interests** None declared.

**Patient and public involvement** Patients and/or the public were not involved in the design, or conduct, or reporting, or dissemination plans of this research.

**Patient consent for publication** Not required.

**Ethics approval** Permission to access, link and analyse these data was granted by the Privacy Advisory Committee (PAC) to the National Health Service (NHS), National Services Scotland (NSS) and the Registrar General.

**Provenance and peer review** Not commissioned; externally peer reviewed.

**Data availability statement** No data are available. This study linked patient information held across administrative health datasets within ISD Scotland, with externally held data held by the Scottish Government and National Records of Scotland. Linkage and de-identification of data was performed by the Information Services Division (ISD) of NHS National Services Scotland (NSS). A data processing agreement between NHS NSS and University of Glasgow was drafted. The University of Glasgow were authorised to receive record linked data controlled and held by ISD within NSS, via access through the national safe haven. The ISD Statistical Disclosure Control Protocol was followed. It is therefore not possible to share data with other parties.

**ORCID iDs**
Laura Anne Hughes-McCormack http://orcid.org/0000-0001-9498-8045
Ruth McGowan http://orcid.org/0000-0002-2188-2239
Angela Henderson http://orcid.org/0000-0002-6146-3477
Sally-Ann Cooper http://orcid.org/0000-0001-6054-7700

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
