## [Reviewer comments · BMJ Open]

ARTICLE DETAILS

TITLE (PROVISIONAL)	Birth incidence, deaths, and hospitalisations of children and young people with Down syndrome, 1990-2015: birth cohort study
AUTHORS	Hughes-McCormack, Laura; McGowan, Ruth; Pell, J. P.; Mackay, Daniel; Henderson, Angela; O'Leary, Lisa; Cooper, Sally-Ann

VERSION 1 – REVIEW

REVIEWER	Henrik Hasle Paediatrics, Aarhus University Hospital, Denmark
REVIEW RETURNED	22-Sep-2019

GENERAL COMMENTS	The authors describe the incidence of Down syndrome, hospitalisation and death rate among 1,479 children/young people with Down syndrome from 1990-2015 in Scotland. The conclusion of the abstract is a very long general discussion without any real facts or conclusions. The abstract fails to comment whether the incidence of Down syndrome has changed over time. On page 9 it's mentioned that the incidence appears to have risen. What is the statistics on this statement? It's interesting that prenatal diagnosis in Scotland does not lead to a reduction of DS birth in contrast to some other countries, e.g. Denmark where the termination rates are high (>90%) if DS is diagnosed (see e.g. Lou et al. Acta Obstet Gynecol Scand. 2018;97:195). A comment on this marked difference between countries would be interesting. Since the data are from cytogenetic registries it should be possible, and of interest to report the frequency of sole trisomy, Robertsonian translocation, and mosaicism. Did the mortality change with cytogenetic group as reported previously (e.g. Zhu et al. Genet Med 2013;15:64)? Death is given as a total fatality rate during the study which is of limited information. Should be calculated as a death rate by a given age of the Down syndrome cohort. Malignancy is not mentioned among the causes of death. It would be of interest due to the unique cancer profile in DS (e.g. Hasle et al. Genet Med 2016;18:1151). Admissions were more likely to have been emergency admissions (77.6% compared to 71.9%), the difference seems too small to have any clinical consequences.
--

	Admission rate increased over time for the Down syndrome group (from 75.4% to 81.2%), and decreased for the control group (from 63.3% to 43.1%). What is the unit of time?
--	--

REVIEWER	Ania Zylbersztejn University College London Great Ormond Street Institute of Child Health , Population, policy and practice
-----------------	--

REVIEW RETURNED	30-Sep-2019
-------------

GENERAL COMMENTS	Thank you so much for asking me to review this paper, which I read with great interest. The study estimated incidence of Down syndrome in Scotland for the first time (estimated to be 1/1000 births, comparable with reported incidence in England & Wales), and described mortality and healthcare use in children with Down syndrome compared to peers. The authors use novel linkage between test results from All NHS Scottish Regional Genetics Centres and other administrative records maintained by the National Services Scotland. This data can provide important insights into incidence of Down syndrome in Scotland and healthcare needs of children with Down syndrome. However, the paper lacks some important detail in the methods section, and I have a few questions about how missing data was handled and how hospital admissions were defined, as this will have significant implications on results. I would recommend major revisions prior to considering this paper for publication. Major comments:  1. How were hospital admissions defined – were transfers between hospitals recorded in SMR01 treated as one or separate admissions? Would a transfer to Neonatal Intensive Care Unit after birth be counted as the first hospital admission? Hospital transfers should be linked into continued periods of hospital care to better account for differences in length of hospital stays and compare more fairly risk of first admission after birth and readmission after. 2. It is unclear how the authors accounted for missing data. The authors mention that there was missing data on SIMD, CHI and sex and this made matching not possible for 188 children with Down syndrome. These children with missing data should be excluded from all analyses (Cox PH models as well as any derived % of children with Down syndrome) to ensure the same cohort is compared in adjusted and unadjusted models. This is especially important when missing CHI: if I understand correctly, if children were missing CHI, they had no link to death and hospital record (as stated on p6-7) so it is unknown if they experienced either of these outcomes. These children should be excluded from calculating the % of children with Down syndrome who died or had a readmission as there's no follow-up for outcomes. 3. The methods section is lacking some key details. The paper would be clearer if the authors had a variables section first, defining outcomes of interest (death, first admission after birth, length of this admission, readmission after?), and specifying all covariates used and describing how these were derived. E.g, could you provide additional information about SIMD indicator? Is it based on postcode at birth, or a different area-level measure? Was matching done based on postcode or SIMD quintile? What descriptive data on risk of admission was derived (p7)? This section can then be followed by analysis section detailing all work described in results section as some analyses are not described
--

	(e.g. info derived in table 2, deaths by underlying cause of death etc.) Minor comments: Methods: 4. Study sample, setting, process (p6): is the screening done during pregnancy, and if so what is the uptake of the screening? How likely is it that some infants with Down syndrome are first diagnosed at birth and therefore missed from the study? 5. Follow-up (p7): you could clarify earlier on that children were followed up from birth, and all models were censored on death or 1st December 2015 (whichever came first) unless stated otherwise. 6. Duration of hospitalisation (p8): the authors state that data is skewed (hence use of log-transformation) so median and non-parametric tests are recommended to compare length of first admission after birth (rather than mean and t-test) 7. For all of Cox PH models, it might be clearer to say that you compared relative risks of each event using hazard ratios rather than that you analysed time to event. Results: 8. Paragraph 1 (p1): How many children had complete information on CHI, SIMD & sex? This complete case cohort should be the basis of your all analyses that compare Downs to matched peers. What was the distribution of births by SIMD? 9. Figure 1: perhaps you could add a smoothed trend line in your plot of incidence of Down syndrome over time (e.g. moving average? Or lowess smoother?) to account for noise from year on year variation 10. Mortality (p10): why was mortality rate not calculated as per 1000 person years to account for differences in follow-up and age-at-death in Down syndrome group and matched controls? Could you provide some measure of median age at death in the two groups or % of deaths in infancy? The proportion of children who died should be based only on children who had a CHI number, otherwise some have unknown outcome? 11. Admissions (p11): the proportion of children with admissions should be based on those who have a link to hospital record. Instead of reporting the chi square statistic/t-test statistics and number of degrees of freedom, it is more useful to see the actual measures (e.g. age at admission etc) and p-value 12. Duration of hospitalisation (p13) – what does coef mean? If it's a log-transformed duration, could you exponentiate it to reflect length in days? Same in table 5 13. All tables should have abbreviations explained in legend (CI, SIMD, HR) like in table 1 14. Risk of readmission (p14) – shouldn't this be based on the number of children who had one hospital admission i.e. 1159 children with Down's syndrome and 3,351 children with no Down syndrome, rather than the total sample? They had to have an admission to be at risk of readmission. Please also clarify the sentence "when adjusted for sex..." 15. For the risk of readmission, could the authors provide a measure of average age at start of follow-up? Age at baseline could also be an important confounder to include in this model 16. All tables: is there an explanation for why the risk of all cause hospital admission, emergency admissions was 90% lower in the least recent cohort vs most recent cohort? Is it due to differences in length of available follow-up?
--	---

	Discussion 17. The authors should be careful with statements about poor quality management of health care at the primary care level. High rates of readmissions compared to peers could reflect complex mix of other factors, including presence of other comorbidities that the authors did not account for. 18. The discussion of effect of SIMD – wouldnt lack of SES gradient in first hospital admissions be a positive sign that children with Down syndrome receive comparable hospital care irrespective of what SES quintile they were born in? How were births distributed by SIMD?
--	--

REVIEWER	Georgina Williams University Hospitals Wales - Noah's Ark Children's Hospital
REVIEW RETURNED	09-Oct-2019

GENERAL COMMENTS	Thank you for this important and interesting paper. This paper provides key information regarding the population of children with DS in Scotland. In particular the incidence of Down syndrome live births and also death rates. The results of this paper will be vital for families, medical professionals, researchers and for service planning. The study has many strengths including the completeness of its data over a long time period. The matched control group and adjustments for age, gender, SIMD have also added strength to this study. I have made comments to the attached paper but overall I felt that although mentioned in the introduction the medical conditions known to be associated with DS were not adequately represented considered when discussing differences seen in hospital admissions between those with DS and the general population. Underlying cardiac conditions, immunological differences, gastrointestinal conditions and intellectual disability are likely to account for most of the differences seen in admissions. You mention that higher rates of admissions and re-admissions are suggestive of poorer quality of management at primary care level but instead I would say it is more likely to be related to their underlying medical conditions. I suspect these patients do not see their GPs very often but are managed more by community paediatricians and specialists including cardiologists, gastroenterologist, physiotherapists etc. The lack of data regarding underlying conditions should be included in the limitations as the control population was not matched for these.
--

VERSION 1 – AUTHOR RESPONSE

Reviewers' Comments to Author:

Reviewer: 1

On page 9 it's mentioned that the incidence appears to have risen. What is the statistics on this statement?

*Response: We acknowledge the year to year variation, and have toned down and amended this comment:

“Table 1 and figure 1 show the incidence of Down syndrome live-births for each year of the study period (with a moving average smoothed trend line to account for noise from year on year variation). The birth rate in Scotland has fallen since the early 1990s, whilst the incidence of Down syndrome shows some year-to-year variation and appears to have remained the same or risen since the early 1990s.”

The conclusion of the abstract is a very long general discussion without any real facts or conclusions. The abstract fails to comment whether the incidence of Down syndrome has changed over time.

*Response: We have added into the abstract that there has been no reduction in incidence over time.

It's interesting that prenatal diagnosis in Scotland does not lead to a reduction of DS birth in contrast to some other countries, e.g. Denmark where the termination rates are high (>90%) if DS is diagnosed (see e.g. Lou et al. *Acta Obstet Gynecol Scand.* 2018;97:195). A comment on this marked difference between countries would be interesting.

*Response: We agree with the reviewer, so have added to the discussion (p.23):

“Previous research in Scotland showed that decisions to terminate pregnancy following antenatal diagnosis of aneuploidy had fallen over time,¹ which our results are possibly also suggestive of from the early 1990s to more recently. In contrast, in some other countries (e.g. Denmark), following the introduction of antenatal combined screening for Down syndrome in 2004, there was a concurrent increase in prenatal diagnoses and the number of Down syndrome live births decreased suddenly and significantly before then stabilising. A higher proportion of the Scottish population have catholic faith compared with Denmark which might contribute in part to differences between countries: a range of complex factors influence women's decision-making regarding prenatal screening and diagnosis of Down syndrome,”

Since the data are from cytogenetic registries it should be possible, and of interest to report the frequency of sole trisomy, Robertsonian translocation, and mosaicism.

Did the mortality change with cytogenetic group as reported previously (e.g. Zhu et al. *Genet Med* 2013;15.64)?

*Response: We agree this would be interesting, but in view of the small numbers of deaths (92), and that the majority of the children/young people had Down syndrome due to trisomy 21, with mosaicism being particularly uncommon, we are unable to analyse deaths by cytogenetic group.

Death is given as a total fatality rate during the study which is of limited information. Should be calculated as a death rate by a given age of the Down syndrome cohort.

*Response: We agree with the reviewer. A new body of data reporting death rates by age group has been added to the death results (p.12):

“Death rates per 1,000 of the population by age groups for the Down syndrome population versus controls were; 8.5 (n=11) versus 0 for the 1st month of life, 45.1 (n=53) versus 1.95 (n=12) from 1 - 12 months, 19.4 (n=18) versus 1.21 (n=6) from 13 - 60 months, 10.8 (n=7) versus 0.85 (<5) from 61 - 120 months, 7.59 (n=<5) versus 0 from 121 - 180 months and 0 versus 1.82 (<5) from 181 - 240 months.”

Malignancy is not mentioned among the causes of death. It would be of interest due to the unique cancer profile in DS (e.g. Hasle et al. *Genet Med* 2016;18:1151).

*Response: We agree with the reviewer. The proportion of deaths from malignancy were found for <5 people with Down syndrome so subject to statistical disclosure; but a comment has been added on this (p.12):

“Although less common than some other causes, leukaemia was a cause of death among people with Down syndrome (n=<5) compared to controls where no deaths were from leukaemia.”

Admissions were more likely to have been emergency admissions (77.6% compared to 71.9%), the difference seems too small to have any clinical consequences.

*Response: We agree it is small difference, but it was statistically significant (as reported).

Admission rate increased over time for the Down syndrome group (from 75.4% to 81.2%), and decreased for the control group (from 63.3% to 43.1%). What is the unit of time?

*Response: We have added the missing information in the abstract and discussion that the unit of time was from 1990-2003 to 2004-2015.

Reviewer: 2

Major comments:

1. How were hospital admissions defined – were transfers between hospitals recorded in SMR01 treated as one or separate admissions? Would a transfer to Neonatal Intensive Care Unit after birth be counted as the first hospital admission? Hospital transfers should be linked into continued periods of hospital care to better account for differences in length of hospital stays and compare more fairly risk of first admission after birth and readmission after.

*Response: We have added more detail about how hospital admissions were defined (p.7):

“In SMR01, continuous periods of care are accounted for with a Continuous Inpatient Stay (CIS) marker. This CIS marker ensures a series of individual episodes over an unbroken period of care (e.g. transfers between wards or hospitals) can be identified and treated as one admission rather than several admissions. Transfer of a baby to a neonatal intensive care unit after birth is counted as a first admission.”

2. It is unclear how the authors accounted for missing data. The authors mention that there was missing data on SIMD, CHI and sex and this made matching not possible for 188 children with Down syndrome. These children with missing data should be excluded from all analyses (Cox PH models as well as any derived % of children with Down syndrome) to ensure the same cohort is compared in adjusted and unadjusted models. This is especially important when missing CHI: if I understand correctly, if children were missing CHI, they had no link to death and hospital record (as stated on p6-7) so it is unknown if they experienced either of these outcomes. These children should be excluded from calculating the % of children with Down syndrome who died or had a readmission as there's no follow-up for outcomes.

*Response: We agree with the reviewer. The deaths and admissions analyses have been revised with the exclusion of the children with Down syndrome with no matched controls, and the methods and results clarify this and provide information on the reasons for non-matching.

3. The methods section is lacking some key details. The paper would be clearer if the authors had a variables section first, defining outcomes of interest (death, first admission after birth, length of this admission, readmission after?), and specifying all covariates used and describing how these were derived. E.g, could you provide additional information about SIMD indicator? Is it based on postcode at birth, or a different area-level measure? Was matching done based on postcode or SIMD quintile? What descriptive data on risk of admission was derived (p7)? This section can then be followed by

analysis section detailing all work described in results section as some analyses are not described (e.g. info derived in table 2, deaths by underlying cause of death etc.)

*Response: We agree with the reviewer, and have reorganised the methods section, and added the additional information.

Minor comments:

Methods:

3. Study sample, setting, process (p6): is the screening done during pregnancy, and if so what is the uptake of the screening? How likely is it that some infants with Down syndrome are first diagnosed at birth and therefore missed from the study?

*Response: All screening for Down syndrome diagnosis was done postnatally so infants diagnosed after being born would not have been missed. This information has been added now for clarification (p.6).

“All NHS Scottish Regional Genetics Centres (the East, North, South-East and West Scotland Centres) identified all live-birth infants screened positive for Down syndrome postnatally”

4. Follow-up (p7): you could clarify earlier on that children were followed up from birth, and all models were censored on death or 1st December 2015 (whichever came first) unless stated otherwise.

*Response: The follow up has now been clarified earlier in the manuscript (p.8).

“All follow up/censoring: Children were followed up from birth, and all models were censored on death or 1st December 2015 (whichever came first) unless stated otherwise.”

6. Duration of hospitalisation (p8): the authors state that data is skewed (hence use of log-transformation) so median and non-parametric tests are recommended to compare length of first admission after birth (rather than mean and t-test)

*Response: We have amended the text/table with median and a non-parametric test for length of first admission (p.14/15).

7. For all of Cox PH models, it might be clearer to say that you compared relative risks of each event using hazard ratios rather than that you analysed time to event.

*Response: The text has been updated accordingly (p.9)

Results:

8. Paragraph 1 (p1): How many children had complete information on CHI, SIMD & sex? This complete case cohort should be the basis of your all analyses that compare Downs to matched peers. What was the distribution of births by SIMD?

*Response: We have added this information on page 10, including the distribution of births by SIMD (p.10).

9. Figure 1: perhaps you could add a smoothed trend line in your plot of incidence of Down syndrome over time (e.g. moving average? Or lowess smoother?) to account for noise from year on year variation

*Response: We thank the author for this suggestion. A moving average smoothed trend line has been added to the plot of incidence and the text has been updated accordingly (p.10).

10. Mortality (p10): why was mortality rate not calculated as per 1000 person years to account for differences in follow-up and age-at-death in Down syndrome group and matched controls? Could you provide some measure of median age at death in the two groups or % of deaths in infancy? The proportion of children who died should be based only on children who had a CHI number, otherwise some have unknown outcome?

*Response: We agree with the reviewer, and have added death rates, and that most occurred in infancy in the Down syndrome group (p.12).

11. Admissions (p11): the proportion of children with admissions should be based on those who have a link to hospital record. Instead of reporting the chi square statistic/t-test statistics and number of degrees of freedom, it is more useful to see the actual measures (e.g. age at admission etc) and p-value

*Response: We agree with the reviewer. The admissions analyses have been revised with the exclusion of the children with Down syndrome with no matched controls. We have reported both statistics and other information such as age at admission (e.g. p.15 & now added for readmissions on p.20).

12. Duration of hospitalisation (p13) – what does coef mean? If it's a log-transformed duration, could you exponentiate it to reflect length in days? Same in table 5

*Response: Coefficient has now been properly labelled (p19). The log-transformed result for the group difference in length of stay has been exponentiated to help make sense of this figure (presented as a ratio).

“This log transformed Coefficient can be interpreted as a ratio of difference for the Down syndrome population compared to controls (exponentiate of the log transformed coefficient, $\exp(.248)=1.28145993219 - 1 \times 100=28$). This means the average length of stay increased by 28% from being in the Down syndrome population, compared to the controls. “

13. All tables should have abbreviations explained in legend (CI, SIMD, HR) like in table 1

*Response: The text has been updated accordingly.

14. Risk of readmission (p17) – shouldn't this be based on the number of children who had one hospital admission i.e. 1159 children with Down's syndrome and 3,351 children with no Down syndrome, rather than the total sample? They had to have an admission to be at risk of readmission. Please also clarify the sentence “when adjusted for sex...”

*Response: The text has been amended accordingly (figures have been changed and the sentence has been clarified).

“Of the Down syndrome group, 963/1,130 (85.2%) had at least one re-admission, compared to 1,700/3,351 (50.7%) of the control group ($\chi^2=132.299$; $df=1$; $P<0.001$).”

15. For the risk of readmission, could the authors provide a measure of average age at start of follow-up? Age at baseline could also be an important confounder to include in this model

*Response: Age at baseline information now added (p.17).

“The baseline age at readmission was lower for the Down syndrome group with a mean age of 1.16 years ($SD=1.79$) compared to the controls with a mean age of 1.44 years ($SD=2.25$; $t(7772)=5.16$, $P<.001$).”

16. All tables: is there an explanation for why the risk of all cause hospital admission, emergency admissions was 90% lower in the least recent cohort vs most recent cohort? Is it due to differences in length of available follow-up?

*Response: We have slightly expanded our interpretation of this in the discussion, but note that it is speculative (p21):

“perhaps reflecting changes in medical practice over time, to keep children out of hospital as far as possible, but admitting for serious problems which the Down syndrome children are more likely to experience.”

Discussion

17. The authors should be careful with statements about poor quality management of health care at the primary care level. High rates of readmissions compared to peers could reflect complex mix of other factors, including presence of other comorbidities that the authors did not account for.

*Response: We agree with the reviewer and have toned this statement down. We have also included in the limitations that we did not aim to investigate underlying medical conditions (p.21, 24).

“Additionally, the high rates of emergency admissions and re-admissions may indicate some poorer quality management of health care at the primary care level (to avoid admissions), and have not to our knowledge been reported before, though other factors, including underlying health conditions may contribute.”

“We did not aim to investigate underlying conditions which are associated with admissions and death in the two groups.”

18. The discussion of effect of SIMD – wouldnt lack of SES gradient in first hospital admissions be a positive sign that children with Down syndrome receive comparable hospital care irrespective of what SES quintile they were born in? How were births distributed by SIMD?

*Response: The distribution of births by SIMD has now been added to the results (p.10), and the following added to the discussion (page 22)

“the children with Down syndrome appear to receive comparable care in terms of first admission, regardless of their neighbourhood,.”

Reviewer: 3 (The following are copied and pasted comments the reviewer (3) made directly on the manuscript document)

*Response: We agree with the reviewer and have toned this statement down. We have also included in the limitations that we did not aim to investigate underlying medical conditions (p.21 & 24).

Abstract

Remove full stop after 'first admission'.

*Response: Full stop removed

I am unclear what you mean by "first discharge where appropriate" and I cannot see where this is mentioned again. Please can you clarify this outcome?

*Response: We have moved the description for this into the new variables section so hopefully this is clearer (it was previously described within the analyses section) and is defined as 'discharge type

from first admission (i.e. regular discharge or irregular discharge [such as a patient discharging himself/herself against medical advice or death])' (p.8).

Strengths and limitations

But not of underlying medical conditions associated with DS. Also no mention anywhere of ethnicity - maybe worth putting a sentence somewhere stating why ethnicity is not explicitly mentioned?

*Response: We have added to the third bullet point in the strengths and limitations section with 'underlying conditions were not investigated' (p.3). Also, a sentence was added to the limitations section to highlight that we did not have data on ethnicity (p.24).

"There was no data on ethnicity which is a limitation; the Scottish population is 96% Caucasian."

Introduction

Live birth rate?

*Response: Text updated accordingly (p.4).

Methods

Coded using ICD-10

*Response: Text updated accordingly (p.7)

This is repeated in every section is there a way of summarising or saying that for each of the outcomes DS was entered as a binary.....

*Response: We agree with the reviewer that this section was too repetitive. The text has been updated accordingly (p.9).

Results

There has been considerable variation year on year and the increase is slight?

*Response: yes, we agree, and are now more cautious in our interpretation of this.

This table would be easier to read if 0 was added before the decimal point and if they all just went to two decimal places for consistency. The same for Table 4,5 and 6

*Response: We thank the reviewer for this suggestion. The tables have been updated accordingly.

Discussion

I would put the information about incidence of live births and death rates first.

*Response: This has been revised accordingly (p.21).

This seems to be a big jump as we do not have enough information on the background conditions for these children which are the likely cause of them needing to be admitted to hospital more frequently than the general population. Readmission rates are very likely to be related to chronic conditions. Primary care are unlikely to be involved very much in these children's care and they are managed more by community paediatricians and where there are other underlying conditions such as congenital heart disease (40-60% of those with DS) the relevant specialist.

*Response: We have toned the statements down about management of health at the primary care, and added that we did not aim to investigate medical conditions (p.21 & 24).

Is the population of people with DS large enough that this would introduce inequality? I think this statement seems a bit odd but I do agree with your next sentence that they need support regardless of the area they reside in

*Response: We have removed the comment on inequality.

I think this paragraph should come first in your discussion

*Response: This has been revised accordingly (p.21).

This is really long could it be shortened and other studies referenced?

*Response: The text has been re-organised to make the section easier to read.

Feels like a bit of a jump? There is considerable variation year on year when you look at Figure 1 but I agree a slight trend upwards.

*Response: We are now more cautious in our interpretations, and acknowledge the variation year on year.

Although this may be secondary to co-morbidities associated with DS

*Response: We have added to the limitations that we did not aim to investigate comorbidities (p.24).

Be aware that some families may also feel the benefit of having time in hospital to get support, necessary treatments and even for some respite - up to you but I would perhaps change or remove the word negative.

*Response: We suspect that most families with a child ill in hospital will visit regularly and stay to support their child, bringing stress and disruption to their lives, so do not think it comparable to respite care. We agree with the reviewer that the word negative is best removed, and have done so. (p.24).

VERSION 2 – REVIEW

REVIEWER	Henrik Hasle Aarhus University Hospital, Denmark
REVIEW RETURNED	17-Jan-2020

GENERAL COMMENTS	None
------

REVIEWER	Ania Zylbersztejn University College London Great Ormond Street Institute of Child Health , Population, policy and practice
REVIEW RETURNED	31-Jan-2020

GENERAL COMMENTS	I have read revised version of this paper with great interest. The authors have addressed most of suggestions and the changes improved the manuscript: a clearer description of missing data and how it was handled is included, analyses were updated accordingly; the methods section link covers more detail. It is an interesting study, using novel linkage, and it is definitely of interest
--

to the academic community. But I have concerns about the statistical analyses which I noticed upon reading the revised manuscript, which need to be addressed prior to publication, in particular with regards to definition of follow-up time and accounting for differences in length of follow-up time for study subjects in analyses. Further major revisions are required prior to publication.

Major comments:

1. The authors included a definition of hospital admission and if I understand correctly transfer to NICU is counted in the analyses as the first hospital admission (after birth admission). If that is the case, this should be discussed further. Could the authors compare the % of infants in the two groups whose first admission is transfer to NICU (or provide some stats based on age at admission if NICU is not indicated in the data)? As authors state in the introduction children with Down syndrome have higher risk of comorbidities (such as congenital heart disease) so they are probably more likely to be transferred to NICU and this is an important explanation of increased risk of first hospital admission rates. It also seems unlikely that the age at first admission would be normally distributed and is likely to be skewed towards lower values – the authors should report median and IQR rather than means (in days or weeks?).

2. The key issue that needs addressing/clarifying is the variation in length of available follow-up time. This study covers individuals with follow-up information ranging from 25 years (for those born in 1990) to 0 days (for births on 1st December 2015). Including all of these children makes sense for calculation of incidence of DS (since DS is diagnosed at birth). For analyses of prognostic outcomes (mortality, hospitalisation) the authors should consider having a minimum “potential” follow-up duration that all children have (e.g. min 1 year of follow-up, meaning that the analyses focus on births in 1990-1st Dec 2014. Or minimum 6 months?). That way, all children have some follow-up time when they could potentially experience the outcome.

3. Given the variation in length of follow-up between people in the study and time-to-event nature of data, all descriptive analyses for risks of first admission, readmission and mortality should focus on incidence rates (per person time) rather than % out of all population. Alternatively the authors could ensure the same length of follow-up for everyone, e.g. by looking at proportion with the first hospital admission in e.g. infancy only (and equivalent for other outcomes) that is, allowing for every included birth to have the same potential follow-up until first readmission, 1st birthday or death (whichever occurred first).

For example, when comparing the % of children with at least one hospital admission (i.e. the risk of first admission) born in 1990-2003 and 2004-2015, the authors derive a proportion of children with a hospital admission based on people with 12-25 years of follow-up data vs those who have 0-11 years of follow-up. It is expected, that with longer follow-up, you'll capture more people as you include additional hospital admissions at older ages, but it's not comparing “like-with-like”. Incidence rate or allowing the same length of follow-up for everyone would account for this. Otherwise results are biased.

	Other comments 4. Birth year group, p.8 –least recent cohort should be (1990-2003) rather than (1990-2015) 5. Death rates P 11/12: The authors say in methods that mortality was calculated per person years, but it looks like overall mortality is still reported as % of all individuals (?). Could you express overall mortality rates per 1000 person years (like for rates by age at death), and derive a ratio of these for children with DS vs controls (or you can fit unadjusted Cox PH model to get that) 6. Given large sample sizes, all p-values are statistically significant. I don't see how this (and including test statistics, and degrees of freedom – not common practice) is useful to readers in the main text (the p-values are already included in tables). 7. Age at first admission, p13 & age at readmission p 15 – these variables are likely to be skewed so comparing them between the groups using mean is not correct. Authors should consider reporting median & IQR, or % of children per age category. 8. Same applies to length of stay, for which comparison of means is still a key finding despite the distribution of this variable being skewed (abstract, p13, p15, p16). Since the authors used log-linear regression model, you could maybe report the baseline length of stay in control group (i.e. exponentiated intercept) and say that it was 28% higher in the DS group 9. Tables 3-6 – it is not clear from the tables that the models were adjusted for sex, IMD, birth year group, which the description of methods on page 9 suggests. Why did authors remove results by these categories? Could the authors add to caption what variables these models were adjusted for 10. Table 5 – it would be useful to see exponentiated values of coefficients as that way they can be interpreted (instead of writing out the calculation for just one value in text). 11. Table 6 – the caption should say “Risk of readmission” rather than “time to readmission” 12. Abstract, primary outcome: The primary outcome section should match description of methods in the paper eg: “Hazard ratios comparing risk of first hospitalisation (any and emergency), and readmission for children with DS and matched controls were calculated using Cox PH hazards model, length of hospital stay was compared using log-linear regression model.” What part of analyses does “year at first admission, type of first admission, and first discharge” refer to, is this what models were adjusted for? 13. Abstract, results: instead of citing the coefficient it would be better to say “children with Down syndrome had 28% longer first admission after birth”
--	--

VERSION 2 – AUTHOR RESPONSE

Major comments:

1. The authors included a definition of hospital admission and if I understand correctly transfer to NICU is counted in the analyses as the first hospital admission (after birth admission). If that is the case, this should be discussed further. Could the authors compare the % of infants in the two groups whose first admission is transfer to NICU (or provide some stats based on age at admission if NICU is not indicated in the data)? As authors state in the introduction children with Down syndrome have higher risk of comorbidities (such as congenital heart disease) so they are probably more likely to be transferred to NICU and this is an important explanation of increased risk of first hospital admission rates. It also seems unlikely that the age at first admission would be normally distributed and is likely to be skewed towards lower values – the authors should report median and IQR rather than means (in days or weeks?).

- Unfortunately our dataset does not specifically distinguish neonatal intensive care, so we are unable to perform the additional suggested analyses. We have added the following to the discussion: “Children with Down syndrome have higher risk of comorbidities, so may be more likely to be transferred to neonatal intensive care units which might explain the increased risk of first hospital admission rates, but we could not distinguish such admissions in our dataset.” (page 21). We have added the statistics for age at first admission: “Compared with the control group, the Down syndrome group were younger at the time of first hospital admission (median=0 months [IQR 0-300 months] versus 24 months [IQR 0-288])” (page 13). We have also added incidence rates for first hospital admissions at different ages: “Admission incidence rates per person time (per 1,000) by age groups for the Down syndrome population versus controls were; 72.89 [CI 68.72-77.32] (n=1,105) versus 40.51 [CI 39.15-41.92] (n=3,305) for all ages, 38.87 [CI 26.05-57.99] (n=24) versus 13.29 [CI 9.79-18.06] (n=41) for the first month of life, 56.44 [CI 52.03-61.23] (n=580) versus 24.94 [CI 23.62-26.33] (n=1,298) from 1-12 months, 117.47 [CI 107.62-128.22] (n=501) versus 74.55 [CI 71.32-77.92] (n=1,966) from 13+ months.” (page 13).

2. The key issue that needs addressing/clarifying is the variation in length of available follow-up time. This study covers individuals with follow-up information ranging from 25 years (for those born in 1990) to 0 days (for births on 1st December 2015). Including all of these children makes sense for calculation of incidence of DS (since DS is diagnosed at birth). For analyses of prognostic outcomes (mortality, hospitalisation) the authors should consider having a minimum “potential” follow-up duration that all children have (e.g. min 1 year of follow-up, meaning that the analyses focus on births in 1990-1st Dec 2014. Or minimum 6 months?). That way, all children have some follow-up time when they could potentially experience the outcome.

- Thank you for this suggestion. All children who do not have a minimum of 1 year follow up have been removed from hospital and death analyses. All relevant figures have been revised accordingly. Please see:

Page 7: “Down syndrome infants who could not be matched due to missing data or had less than one year of available follow up time (i.e. births in 2015) were excluded from analyses other than birth rate. The CHI also provides a means to record link each person identified with Down syndrome and their matched controls to routinely collected hospital admissions data (Scottish Morbidity Records 01:

SMR01),¹⁵ and National Records of Scotland death certificate data.¹⁶ The end of the period of follow-up was 1st December 2015. However, children born in 2015 were excluded from the death and hospital admission analyses to allow a minimum follow up time of 1 year for all children.”

Page 10: “A further 57 children were excluded as they were born in 2015 (and had less than 1 year of follow up time).”

3. Given the variation in length of follow-up between people in the study and time-to-event nature of data, all descriptive analyses for risks of first admission, readmission and mortality should focus on incidence rates (per person time) rather than % out of all population. Alternatively the authors could ensure the same length of follow-up for everyone, e.g. by looking at proportion with the first hospital admission in e.g. infancy only (and equivalent for other outcomes) that is, allowing for every included birth to have the same potential follow-up until first readmission, 1st birthday or death (whichever occurred first).

- To take into account the time-to-event nature of the data, all figures have been revised as incident rates (per person time per 1000) relating to risks of first admission (all cause and emergency), readmission and mortality. Please see:

Page 12: “Death incidence rates per person time (per 1,000) by age groups for the Down syndrome population versus controls were; 0.56 [CI 0.45-0.68] (n=92) versus 0.03 [CI 0.02-0.04] (n=23) for all ages, 17.81 [CI 9.87-32.17] (n=11) versus 0 for the 1st month of life, 3.87 [CI 2.96-5.07] (n=53) versus 0.17 [CI 0.09-0.29] (n=12) from 1 - 12 months, 0.36 [CI 2.96-5.07] (n=18) versus 0.02 [CI 0.01-0.05] (n=6) from 13 - 60 months, 0.15 [CI 0.07-0.32] (n=7) versus 0.01 [CI 0.00-0.04] (<5) from 61 - 120 months, 0.09 [CI 0.03-0.31] (n=<5) versus 0 from 121 - 180 months and 0 versus 0.02 [CI 0.00-0.08] (<5) from 181 - 240 months.”

Page 13: “Admission incidence rates per person time (per 1,000) by age groups for the Down syndrome population versus controls were; 72.89 [CI 68.72-77.32] (n=1,105) versus 40.51 [CI 39.15-41.92] (n=3,305) for all ages, 38.87 [CI 26.05-57.99] (n=24) versus 13.29 [CI 9.79-18.06] (n=41) for the first month of life, 56.44 [CI 52.03-61.23] (n=580) versus 24.94 [CI 23.62-26.33] (n=1,298) from 1-12 months, 117.47 [CI 107.62-128.22] (n=501) versus 74.55 [CI 71.32-77.92] (n=1,966) from 13+ months.”

Page 14: “Emergency admission incidence rates per person time (per 1,000) by age groups for the Down syndrome population versus controls were; 56.78 [CI 53.13-60.72] (n=861) versus 28.88 [CI 27.73-30.07] (n=2,352) for all ages, 27.53 [CI 17.12-44.23] (n=17) versus 11.03 [CI 7.88-15.43] (n=34) for the first month of life, 48.66 [CI 44.57-53.11] (n=500) versus 20.42 [CI 19.23-21.69] (n=1,063) from 1-12 months, 80.66 [CI 72.57-89.65] (n=344) versus 47.71 [CI 45.14-50.42] (n=1,255) from 13+ months.”

Page 16: “Re-admission incidence rates per person time (per 1,000) for the Down syndrome population versus controls were 54.85 [CI 51.46-58.46] (n=945) versus 15.06 [CI 14.36-15.80] (n=1695).”

For example, when comparing the % of children with at least one hospital admission (i.e. the risk of first admission) born in 1990-2003 and 2004-2015, the authors derive a proportion of children with a hospital admission based on people with 12-25 years of follow-up data vs those who have 0-11 years of follow-up. It is expected, that with longer follow-up, you'll capture more people as you include additional hospital admissions at older ages, but it's not comparing "like-with-like". Incidence rate or allowing the same length of follow-up for everyone would account for this. Otherwise results are biased.

- Incidence rates are now presented – as described above.

Other comments

4. Birth year group, p.8 –least recent cohort should be (1990-2003) rather than (1990-2015)

- Thank you for pointing this out. It has now been revised.

5. Death rates P 11/12: The authors say in methods that mortality was calculated per person years, but it looks like overall mortality is still reported as % of all individuals (?). Could you express overall mortality rates per 1000 person years (like for rates by age at death), and derive a ratio of these for children with DS vs controls (or you can fit unadjusted Cox PH model to get that)

- This has been revised. Please see page 12: “Death incidence rates per person time (per 1,000) by age groups for the Down syndrome population versus controls were; 0.56 [CI 0.45-0.68] (n=92) versus 0.03 [CI 0.02-0.04] (n=23) for all ages...”.

6. Given large sample sizes, all p-values are statistically significant. I don't see how this (and including test statistics, and degrees of freedom – not common practice) is useful to readers in the main text (the p-values are already included in tables).

- All test statistic information has been removed from the text accordingly.

7. Age at first admission, p13 & age at readmission p 15 – these variables are likely to be skewed so comparing them between the groups using mean is not correct. Authors should consider reporting median & IQR, or % of children per age category.

- Age at first admission & age at readmission are now presented as medians (and IQR).

8. Same applies to length of stay, for which comparison of means is still a key finding despite the distribution of this variable being skewed (abstract, p13, p15, p16). Since the authors used log-linear regression model, you could maybe report the baseline length of stay in control group (i.e. exponentiated intercept) and say that it was 28% higher in the DS group

- Median/IQR are now presented for length of stay (pages. 14/15). The exponentiated intercept has been added (page 15/16)

9. Tables 3-6 – it is not clear from the tables that the models were adjusted for sex, IMD, birth year group, which the description of methods on page 9 suggests. Why did authors remove results by these categories? Could the authors add to caption what variables these models were adjusted for

- Thank you for this suggestion. All tables have now been captioned to indicate they were stratified models by sex, birth year group, SIMD.

10. Table 5 – it would be useful to see exponentiated values of coefficients as that way they can be interpreted (instead of writing out the calculation for just one value in text).

- All exponentiated values of coefficients have been interpreted and presented as the % of increase in time (added in table 5).

11. Table 6 – the caption should say “Risk of readmission” rather than “time to readmission”

- Thank you for highlighting this. This has been revised accordingly for table 6.

12. Abstract, primary outcome: The primary outcome section should match description of methods in the paper eg: “Hazard ratios comparing risk of first hospitalisation (any and emergency), and readmission for children with DS and matched controls were calculated using Cox PH hazards model, length of hospital stay was compared using log-linear regression model.” What part of analyses does “year at first admission, type of first admission, and first discharge” refer to, is this what models were adjusted for?

- The primary outcome section has been revised as per the reviewer's suggestions.

13. Abstract, results: instead of citing the coefficient it would be better to say "children with Down syndrome had 28% longer first admission after birth"

- The results section of the abstract has been revised as per the reviewer's suggestion.

VERSION 3 – REVIEW

REVIEWER	Ania Zylbersztejn University College London Great Ormond Street Institute of Child Health , Population, policy and practice
REVIEW RETURNED	09-Mar-2020
GENERAL COMMENTS	None